# A light-activatable theranostic combination for ratiometric hypoxia imaging and oxygen-deprived drug activity enhancement

Lei Ge [1], Yikai Tang[1], Chongzhi Wang[1], Jian Chen[1], Hui Mao [2] & Xiqun Jiang [1] ✉

While performing oxygen-related tumour treatments such as chemotherapy and photodynamic therapy, real-time monitoring hypoxia of tumour is of great value and significance. Here, we design a theranostic combination for light-activated ratiometric hypoxia imaging, hypoxia modulating and prodrug activation. This combination consisted of an oxygen-sensitive near-infrared-emitting ratiometric phosphorescence probe and a hypoxia-activated pro-drug-loaded covalent organic framework. In this combination, the probe plays two roles, including quantitative monitoring of oxygen concentration by ratiometric imaging and consuming the oxygen of tumour under light excitation by photodynamic therapy. Meanwhile, the enhanced hypoxia micro-environment of tumour can raise the cytotoxicity of prodrug loaded in covalent organic framework, resulting in boosting antitumour therapeutic effects in vivo. This theranostic combination can precisely provide therapeutic regime and screen hypoxia-activated prodrugs based on real-time tumour hypoxia level, offering a strategy to develop hypoxia mediated tumour theranostics with hypoxia targeted prodrugs.

Hypoxia is a hallmark characteristic of the tumour microenvironment (TME) highlighted with poor tissue oxygenation. It plays key roles in tumour initiation, progression, metastasis, and resistance to conventional and targeting cancer therapies[1–4], including immunotherapy[5,6], photodynamic therapy[7,8], radiation therapy[9] and chemotherapy[10]. Understanding, even targeting and controlling, tumour-specific hypoxia is essentially important to develop precision diagnosis and treatment[11,12]. Indeed, a range of hypoxia-responsive theranostic systems have been developed in recent years[13–16]. They typically are consisted of two key components, a fluorescent probe for imaging detection of hypoxia and a therapeutic agent for treatment. However, most hypoxia-responsive probes currently reported still stay in qualitative stage, and lack the quantitative capability[17–20], while the therapeutic agents have low efficacy under the low oxygen environment due to TME induced drug resistance and immunosuppression. The strategy of using hypoxia-activated bioreductive agents, such as

tirapazamine (TPZ)[21–24], which are non-toxic under the normoxia but activated for biological reactions in hypoxia microenvironment[25], resulting in high cytotoxicity to kill cancer cells[26–28] also is attractive with potential to overcome the therapeutic challenges in treating hypoxic tumours. However, the hypoxic cytotoxin, such as TPZ, usually accumulates in vicinity of tumour vessels where the oxygen level is relatively high, and can't diffuse into hypoxic areas far from tumour vessels. Thus, it only can be effective if one can determine the level of hypoxia in each area of tumour or make heterogeneous oxygen distribution in tumour become homogenous hypoxia[29,30].

In conventional light-activated theranostic systems, photosensitizers and therapeutic agents are usually co-loaded into the one nanocarrier, followed by subsequent optical imaging, photodynamic therapy (PDT), or chemotherapy. However, there are still two main issues: (a) the amount ratio of photosensitizer to therapeutic agent cannot be adjusted in real time according to the imaging results,

[1]College of Chemistry and Chemical Engineering, Nanjing University, 210023 Nanjing, China. [2]Department of Radiology and Imaging Sciences, Emory University, Atlanta, GA 30322, USA. ✉e-mail: jiangx@nju.edu.cn

treatment progress and other factors; (b) Unable to immediately switch to another drug for therapy. These issues greatly limit the prospects for clinical application of light-activated theranostic system and also inspired us to construct a theranostic combination with two separated functional units: quantitative hypoxia reporter and hypoxia-activated drugs, to create a maximum treatment response for tumours by selecting favourable drug and therapeutic regime based on hypoxia modulation of tumour.

Here, we report a light-activated theranostic combination having ratiometric imaging and modulation of tissue oxygenation as well as hypoxia-activated prodrug. This rationally designed multifunctional nano-construct combination is comprised of an oxygen-sensitive near-infrared (NIR)-emitting ratiometric phosphorescence nanoprobe (Ir-NP) and a nanoscale covalent organic framework (COF) loaded with hypoxia-activatable prodrug such as TPZ (COF@T) (Fig. 1). Ir-NP, consisting of phosphorescent Ir (IV) metalloporphyrin (IrTBP) and fluorescent porphyrin (TPP) as well as α, β-amino-terminated polyethylene glycol (PEG, $Mw = 2$ kDa), enables measurement of tissue oxygen and photosensitisation for activating encapsulated antitumour prodrug TPZ in low oxygen microenvironment. This theranostic combination not only can quantitatively monitor the tumour hypoxia level by ratiometric NIR imaging, but also can modulate the tumour hypoxia and provide a guide for selecting favourable drug and therapeutic regime to exert the maximum therapeutic efficiency of hypoxia-activated prodrugs.

## Results

### Synthesis and properties of the NIR-emitting ratiometric phosphorescence probe

Ir-tetraphenylbenzoporphyrin (IrTBP), an oxygen-sensitive phosphorescent complex, was initially synthesised (Supplementary Fig. 1), and the structure was confirmed by [1]HNMR and MALDI-TOF-MS spectra (Supplementary Figs. 2 and 3). Ir containing nanoprobes, Ir-NP, were then prepared by self-assembly of chlorosulfonylated IrTBP, chlorosulfonylated tetraphenylporphyrin (TPP) and α, ω-amino-terminated polyethylene glycol in the aqueous solution (Fig. 1 and Supplementary Fig. 4). Dynamic light scattering (DLS) measurement showed that the hydrodynamic diameter of Ir-NP in aqueous solution was about 50 nm and transmission electron microscopy (TEM) observation displayed that Ir-NP had a spherical morphology with an average size of 29.9 nm (Fig. 2a, b and Supplementary Fig. 5). Prepared nanoprobes had great stability in a variety of media evidenced by the colloidal stability measurement (Supplementary Figs. 6 and 7), allowing for adaptation to complex physiological environments.

Ir-NP had a strong Soret band at $\lambda_{max}$ 426 nm and a wide Q band from 500 nm to 650 nm as observed from the UV-Vis absorption of Ir-NP (Fig. 2c). The emission spectra of Ir-NP with signal intensities of peaks at 665 nm and 705 nm from the TPP unit remained unchanged under different oxygen conditions, while the one at 774 nm from the IrTBP unit was oxygen level sensitive, sharply increasing with the decrease of oxygen concentration (Fig. 2d). This is due to the fact that the oxygen can quench the phosphorescence of IrTBP while not affecting the fluorescence of TPP. The ratio of phosphorescence intensities of this peak (774 nm) under the oxygen-free to oxygen-saturated conditions reached 6. All emission peaks of Ir-NP were well resolved without overlapping, which is important for ensuring accurate quantification of the tissue oxygenation level. The mass ratio of chlorosulfonylated IrTBP to chlorosulfonylated TPP was determined as 1/4 to ensure that the ratio of luminescent intensity at 665 nm to 774 nm ($I_{665}/I_{774}$) was suitable for detection (Supplementary Table 1). Importantly, the ratio ($I_{665}/I_{774}$) exhibited an apparent linear

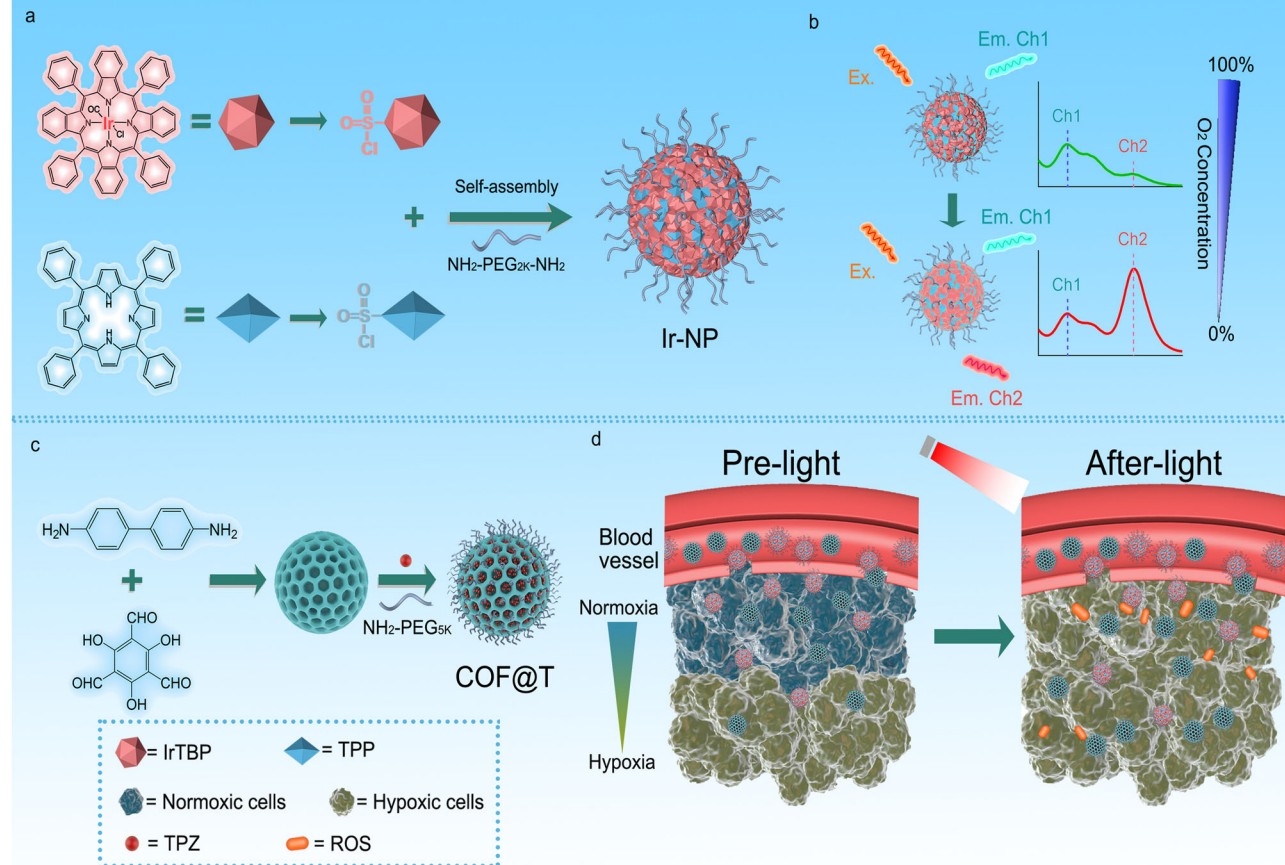

**Fig. 1 | Schematic illustration of Ir-NP based theranostic system. a** Fabrication of Ir-NP. **b** Hypoxia response illustration of Ir-NP. **c** Fabrication of COF@T. **d** The scheme illustrating of the theranostic system, which could improve the therapeutic of hypoxia prodrug (TPZ) upon light irradiation and real-time monitor hypoxia.

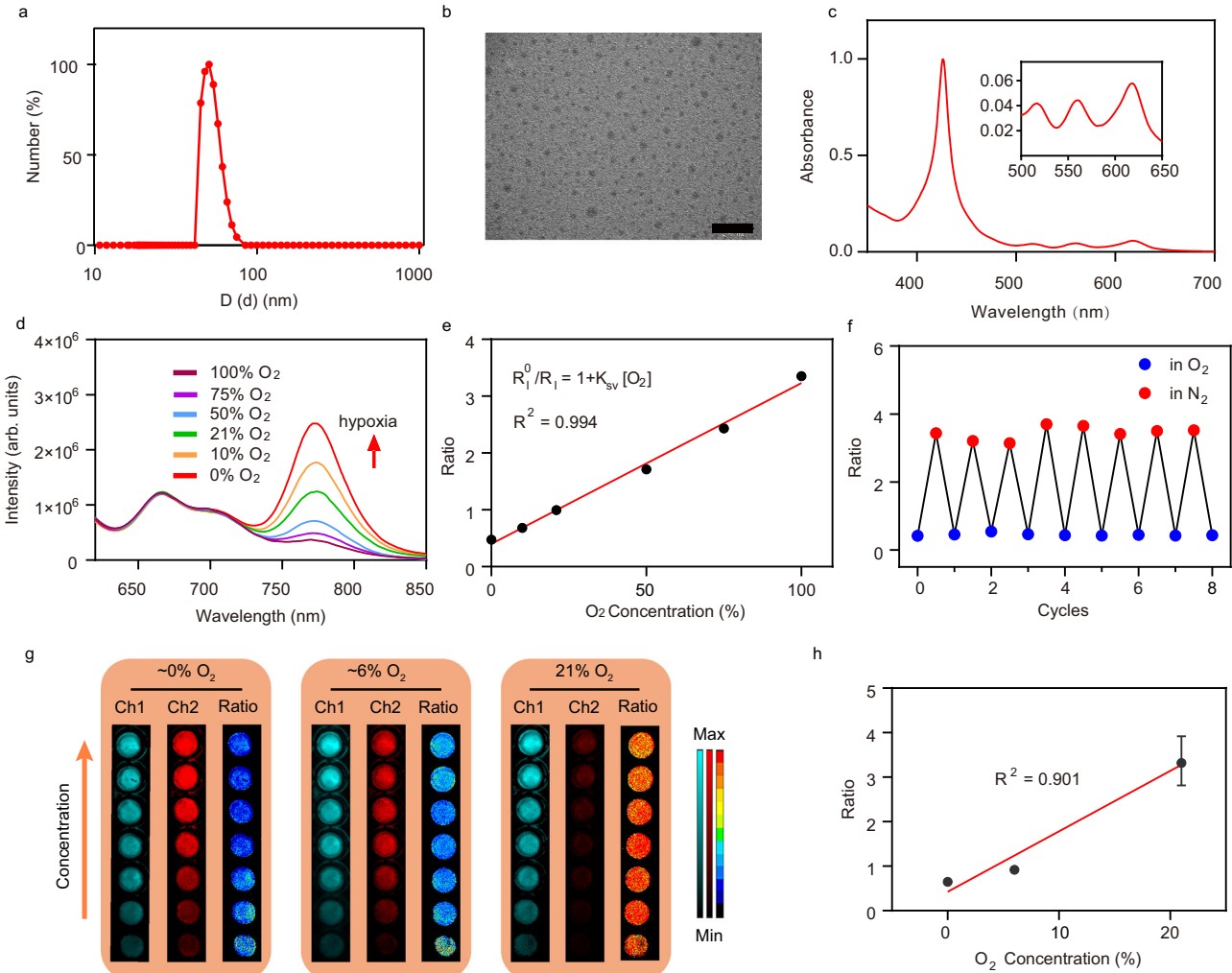

**Fig. 2 | Characterisation of Ir-NP and its ratiometric imaging. a** The hydro-dynamic size distribution of Ir-NP. **b** The TEM image of Ir-NP. Scale bar is 200 nm. ($n$ = 3 independent experiments). **c** The UV-vis absorption of Ir-NP in aqueous solutions. **d** The emission of Ir-NP with different oxygen levels in aqueous solutions, $\lambda_{ex}$ = 620 nm. **e** The linear correlation of the ratio ($I_{774}$ / $I_{665}$) response to oxygen levels of Fig. 1d. **f** Reversible ratio of Ir-NP in nitrogen- and oxygen-saturated aqueous solution. **g** Luminescence imaging and ratio-metric results of Ir-NP with different concentration under different oxygen concentrations. Ratio = Ch1 (675 ± 25 nm)/Ch2 (775 ± 25 nm). **h** The linear correlation of the ratio response to oxygen levels of Fig. 1g. The statistical data are expressed as mean values ± S.D. ($n$ = 7 independent experiments). The term (arb. units) is abbreviated for arbitrary units.

relationship with the oxygen level as the function and the linear correlation coefficient ($R^2$) reached 0.994 (Fig. 2e). The emission peaks and ratio ($I_{665}/I_{774}$) had strong photostability within 10 min irradiation and outstanding pH-resistant durability with a pH range from 2 to 12 (Supplementary Fig. 8). when bubbling $N_2$ and $O_2$ into the aqueous solutions of Ir-NP iteratively, $I_{665}/I_{774}$ displayed the oxygen-responsive changes from the oxygen concentration of 0–100% during eight cycles (Fig. 2f), indicating oxygen-responsive robust of Ir-NP.

Next, we evaluated the capability and performance of using Ir-NP as the oxygen responsive probe under different oxygen conditions (0%, 6%, and 21% $O_2$) in the aqueous solution while varying Ir-NP concentrations (0.1, 0.2, 0.5, 1, 2, 5 and 10 mg/L). The luminescent images of Ir-NP at Chanel 1 (Ch1, range from 640 to 690 nm) and Chanel 2 (Ch2, range from 750 to 800 nm) and the corresponding ratios of the signal intensities from Ch1 and 2 (Ch1/Ch2) were shown in Fig. 2g and Supplementary Fig. 9 revealed that images of Ch1/Ch2 was independent of probe concentration. Moreover, Ch1/Ch2 ratio and the corresponding oxygen concentration exhibited a tight linear relationship with the correlation coefficient ($R^2$) of 0.901 (Fig. 2h), suggesting that Ir-NP is a suitable ratiometric imaging probe for real-time monitoring the oxygen concentration non-invasively and quantitatively.

## Hypoxia responsiveness of the probe in ratiometric imaging of cell

We examined the uptake of Ir-NP by mouse breast 4T1 cancer cells using flow cytometry and confocal laser scanning microscopy (CLSM). The increased intracellular luminescent signal after incubation with Ir-NP was observed as a function of the incubation time (Supplementary Fig. 10). The CLSM images revealed that Ir-NP was mainly distributed in the cytoplasm (Supplementary Fig. 11). When the culturing time of 4T1 cells with probe was constant, the intensity of both Ch1 and Ch2 increased as the nanoprobe concentration elevated. However, the Ch1/Ch2 ratio remained almost the same, independent of Ir-NP concentration (Fig. 3a). Interestingly, as seen in Fig. 3a, b, the length of culturing time and the ratio of Ch1/Ch2 had a strong correlation, suggesting that the ratio of Ch1/Ch2 of Ir-NP can sensitively reflect the oxygen consumption of cells, that is, increasing culturing time of cells increases oxygen consumption. Thus, the change in intracellular oxygen level can be detected by the ratiometric imaging of the emission peaks of Ir-NP. In addition, Ch1/Ch2 ratio becomes more independent on probe concentration at low oxygen content (Fig. 3b), highlighting the merit of our ratiometric hypoxia probe.

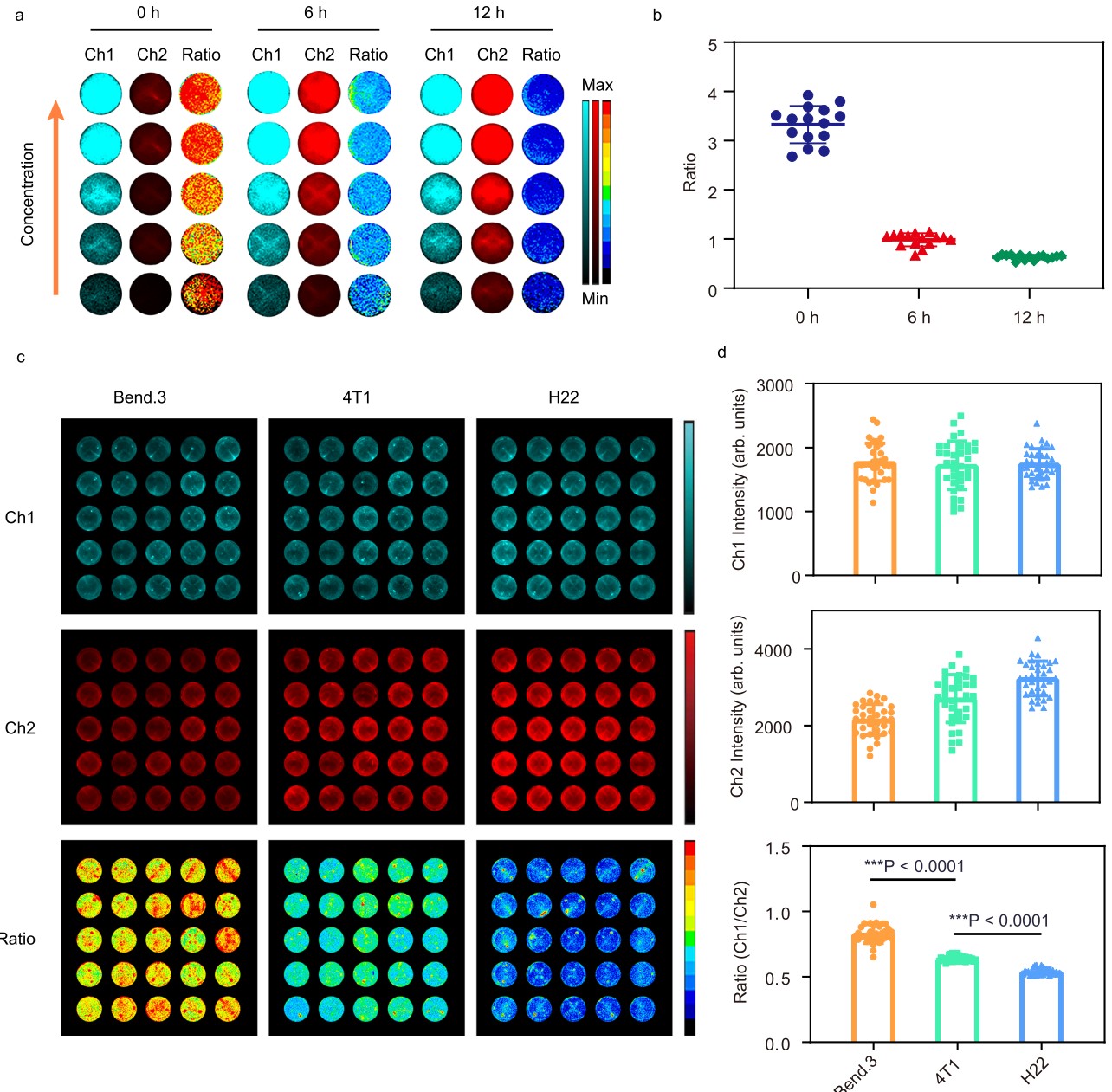

**Fig. 3 | The ratiometric imaging of Ir-NP in vitro. a, b** Luminescent imaging and ratio-metric results of Ir-NP with 4T1 cells after different incubation time. Ratio = Ch1 (650–700 nm) / Ch2 (750 – 800 nm). The statistical data are expressed as mean values ± S.D. (*n* = 15 independent experiments). **c, d** Luminescent imaging and ratio-metric results of Ir-NP with different cells after same incubation time. Ratio = Ch1 (650–700 nm)/Ch2 (750–800 nm). ***P < 0.0001 (Bend.3 vs 4T1), ***P < 0.0001 (4T1 vs H22). One-way ANOVA with Dunnett's multiple comparisons test. The statistical data are expressed as mean values ± S.D. (*n* = 34 independent experiments). The term (arb. units) is abbreviated for arbitrary units.

Then, Ir-NP was used to determine the oxygen consumption rates of different cells, i.e., murine brain microvascular epithelium Bend.3 cells, murine breast cancer 4T1 cells, and murine hepatic cancer H22 cells. After incubation of the cells with Ir-NP for 12 h, the signal intensity of Ch1 was almost same in all three types of cells while the intensities of Ch2 varied in different cells, increasing in the order of Bend.3 < 4T1 < H22 (Fig. 3c and d). Accordingly, the order of Ch1/Ch2 ratios of these cells were Bend.3 > 4T1 > H22, indicating that different cells have different oxygen demands. Compared with the two tumour cell lines (4T1 and H22), the normal cells, Bend.3, showed the lowest oxygen consumption rate. In addition, different cancer cells also exhibited the different level of oxygen consumption as H22 cells showed a lower Ch1/Ch2 ratio than that of 4T1 cells.

## Ratiometric imaging of tumour hypoxia in mice

We subsequently investigated real-time in vivo ratiometric imaging of tumour hypoxia in the BALB/c mice bearing the 4T1 xenografted tumour using the Ir-NP probe. After intravenous injection of Ir-NP (10 mg/kg), the whole-body imaging at the wavelength of Ch1 and Ch2 was performed at different time points post-injection. The biodistribution of the Ir-NP probe in blood, tumour, and major organs from 1 h to 72 h post-injection was also measured (Supplementary Figs. 12–14), indicating probes mainly accumulated in liver and kidney with the blood half-life time to be 5.8 ± 0.12 h. The probe accumulated in tumours reached the maximum at 16 h post-injection and was measured to be 8.5% of injection dose per gram of collected organs (ID g$^{-1}$). We also obtained ex vivo NIR imaging results of major organs

and tumours at 72 h post-injection (Supplementary Fig. 15) and the results showed that the both tumour and liver yielded a strong signal in Ch1 imaging, while only tumour had strong signal in Ch2 imaging, proving the effective tumour accumulation and hypoxia-specific response of Ir-NP probe. Signal intensities of both Ch1 and Ch2 observed in the tumour regions increased gradually from 4 to 16 h after injection and then decreased from 16 to 48 h (Fig. 4a, b). Using the hypoxia-insensitive Ch1 images as a reference, hypoxia-sensitive images from Ch2 imaging yielded improved contrast in delineating the

location and contour of the tumour. Since ratiometric imaging with a hypoxia-insensitive channel as a stable internal reference, the change of the hypoxia-sensitive signal from the channel 2 is not affected by the variation of the probe concentration. This is a substantial specificity advantage over conventional molecular probes as their readout for a specific physiological or biological event is typically dependent on multiple variables, including the probe concentration. In this case, the relatively stable hypoxia level observed over the 72 h post-injection (Fig. 4c) suggests that the hypoxia level at the tumour site appear to be

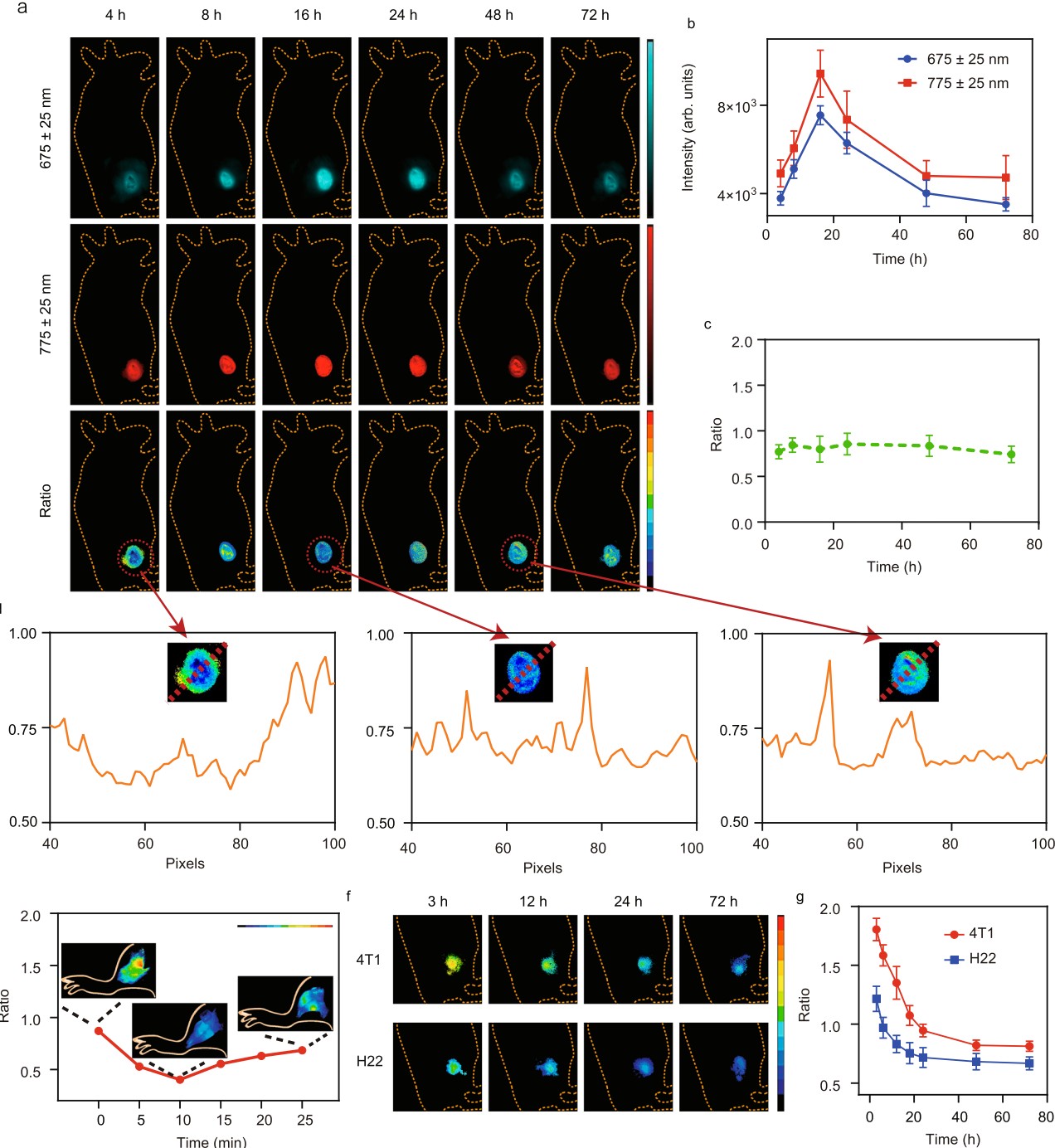

**Fig. 4 | The ratiometric imaging of Ir-NP in vivo. a** The whole-body optical imaging and ratiometric results of mice subcutaneously implanted 4T1 tumours after i.v. injection of Ir-NP. The signal intensity of 675 ± 25 nm and 775 ± 25 nm (**b**) and the ratio (**c**) at different time point. The ratio = *I* (675 ± 25 nm)/*I* (775 ± 25 nm). The statistical data are expressed as mean values ± S.D. (*n* = 5 biologically independent animals). **d** The distribution of the values of tumour ratio imaging along the red dotted line. **e** The real-time ratio-metric changes and representative images of tumour. The ratio-metric imaging (**f**) and results (**g**) of mice subcutaneously 4T1 and H22 cells after i.v. injection of Ir-NP. The term (arb. units) is abbreviated for arbitrary units.

stable once cancer cells proliferate to form a tumour. Importantly, we observed significant heterogeneity of the tissue oxygenation level in the tumour by the ratiometric imaging (Fig. 4d), while it was difficult to obtain based on single fluorescence or phosphorescence imaging. This inhomogeneous tissue oxygenation level within the tumour has been reported as a significant contributor in limiting the therapeutic effect of many cancer therapies, including hypoxia prodrug[31–33].

To examine and validate whether Ir-NP can be used for dynamically monitoring change of the oxygen concentration in tumours, we imaged tumour area after tying the blood vessel to restrict the blood flow to the tumour. Ratiometric imaging with Ch1/Ch2 showed a significant drop in the oxygen level of the tumour from 0 to 10 min after the blood flow was restricted (Fig. 4e). As blood flow resumed, the oxygen level in the tumour gradually increased in next 15 min with the signal ratio of Ch1/Ch2 ascended, indicating that our Ir-NP probe has fast responses to the tissue oxygenation with a high temporal resolution to the alterations of oxygen. When different types of cancer cells ($5 \times 10^6$) were subcutaneously injected into the mice, we also observed the gradual increase of the hypoxia level over time of 3–72 h with hepatic cancer H22 cells exhibiting more rapid oxygen consumption than 4T1 cells (Fig. 4f, g), which is also consistent with the results from our experiments using cell culture assay and the previous report on cellular oxygen consumption rate[34–36]. Subsequently, we substantiated the significantly higher degree of hypoxia in H22 xenografted tumour compared to 4T1 xenografted tumour through the utilisation of oxygen concentration measurement electrodes and pimonidazole staining (Supplementary Fig. 16). The results of hypoxia-inducible factor-1α (HIF-1α) immunohistochemistry staining were also in accordance with this finding (Supplementary Fig. 17). This result indicates that the hypoxia level of tumour area continuously varies before the formation of stable tumour since the proliferation of cancer cells is oxygen consumption.

### Photodynamic therapy of Ir-NP

With a porphyrin component in the Ir-NP structure[37–39], it can function as a photodynamic agent to generate reactive oxygen species (ROS) under light irradiation using a 650 nm laser with singlet oxygen sensor green (SOSG) as probe (Supplementary Fig. 18). With the decrease in $O_2$ concentration, the efficiency of generating singlet oxygen ($^1O_2$) decreased notably (Fig. 5a). Furthermore, we found a stronger intracellular fluorescence from the ROS indicator, DCFH-DA, in 4T1 cells at a higher oxygen concentration under the light irradiation (Fig. 5b). When increasing the concentration of Ir-NP incubated with 4T1 cells from 1 to 15 mg/L and applying a light irradiation to 4T1 cells, the values of Ch1/Ch2 ratio reduced from 3.11 to 1.44 (Fig. 5c) and the number of living 4T1 cells also significantly decrease (Supplementary Fig. 19), meaning the intracellular oxygen concentration significantly decreases and ROS increase after the light irradiation, demonstrating that not only Ir-NP probes can be used as a ratiometric imaging probe to monitor the change of tissue oxygenation in the photodynamic therapy (PDT) but also can be used a hypoxia-targeted and activated photosensitizer for PDT.

### Theranostic combination of Ir-NP and TPZ

Building on the hypoxia-specific ratiometric imaging and photosensitizer functions of Ir-NP probes, we further designed and prepared a covalent organic framework (COF) made of benzidine (BD) and triformyl-trihydroxybenzene (TP), as previously reported by us[40] with hypoxia activatable prodrug TPZ loaded (COF@T) as illustrated in Supplementary Fig. 20. DLS data and TEM images showed that the COF@T had size of about 100 nm and a regular spherical morphology (Supplementary Fig. 21). The BET surface area measurement of COF was found to be 580.42 m² g⁻¹, indicating a highly porous structure. Additionally, the pore size of COF was determined to be 2.4 nm (Supplementary Fig. 22). The loading capacity of TPZ in COF@T was

calculated as 12.1% and the TPZ release from COF@T is shown as Supplementary Fig. 23. Cell toxicity evaluation based on the MTT assay using 4T1 cells and 5 mg/L Ir-NP or different concentrations of COF@T revealed that COF@T had un-detectable cytotoxicity to 4T1 cells under the normoxia and irradiation conditions but exhibited high cytotoxicity under hypoxia conditions with light irradiation at 650 nm (Fig. 5d and Supplementary Fig. 24). The cell viability was significantly lower in the presence of COF@T than the control condition after irradiation, demonstrating that light induces the enhanced cytotoxicity of COF@T (Fig. 5d). Similar results were also obtained from flow cytometry experiments (Fig. 5e). Reassuringly, the theranostic combination exhibited remarkably low cytotoxicity to healthy cells under normoxia conditions without light irradiation (Supplementary Fig. 25).

### Antitumour efficacy of Ir-NP and TPZ theranostic combination

To investigate the in vivo antitumour effect of this theranostic combination, the biodistribution of COF@T in 4T1 tumour-bearing mice was measured firstly (Supplementary Fig. 26 and 27). The blood half-life time of COF@T was determined to be 2.3 ± 0.29 h, longer than that of TPZ (0.8 ± 0.15 h). Besides, the results showed that at 3 and 6 h post-injection, the TPZ level at the tumour site of the COF@T group was higher than that of the free TPZ group by 3.3 and 5.0-fold, respectively. Contributed to the nano structure, COF@T prolonged the blood circulation and enhanced accumulation of TPZ in tumour. Mice bearing subcutaneous 4T1 tumours were divided into two groups for testing the antitumour efficacy of the theranostic combination using two different treatment regimens (Fig. 6a). One group was given one 15-min light irradiation after Ir-NP and COF@T were co-administrated intravenously into the mice with the dose of 20 mg/kg for Ir-NP and 9 mg/kg for TPZ (Regimen-1), while the other group received additional two times of 5-min irradiation on alternate days (Regimen-2). Changes in the tumour oxygenation level during the treatment were monitored by hypoxia-specific ratiometric imaging using the Ir-NP probe. As shown in Fig. 6b, the tissue oxygenation level of the tumours in both groups decreased after the light irradiation on Day 1. However, the tumour oxygenation level in the group treated with Regimen-1 slowly increased over the next few days and recovered into the original level prior to the irradiation treatment. In contrast, the tumour tissue oxygenation level measured from the Regimen-2 group fluctuated in responding to the light exposure on Days 4 and 6, suggesting that the tumour tissue oxygenation level of mice in the Regimen-2 group is dependent on the fractionated PDT. The lower tumour tissue oxygenation level of Regimen-2 group than that of Regimen-1 group led to much large difference in the therapeutic response of COF@T between two groups (Fig. 6c). The concentration changes of TPZ at the tumour site of mice from two groups during the 6-day regimen are shown in Supplementary Fig. 28. Notably, the result suggests that TPZ consistently maintains a high concentration throughout the duration of treatment, thereby ensuring the successful implementation of therapeutic strategy.

We also compare the treatment response in mice treated with PBS, COF@T, and Ir-NP with light (Ir-NP + $h\nu$) (Fig. 6c, d). Mice treated with COF@T or Ir-NP + $h\nu$ exhibited about 80% tumour inhibition effect compared to mice treated with PBS only. In contrast, Regimen 1 and 2 treatments with theranostic combination and light irradiation displayed the better antitumour efficiency. Moreover, compared with single light irradiation treatment, the group received multiple light irradiation treatments showed multi-administration effect, responding with continuous shrinkage of the tumour volume with increasing radiation times. The measurement of tumour weight after sacrificing animals confirmed the effect of tumour inhibition (Supplementary Fig. 29). Furthermore, no significant change in the body weight of those tumour-bearing animals was found over the time of treatment (Supplementary Fig. 30). H&E staining revealed no aberrations of all primary organs (Supplementary Fig. 31), suggesting a good biocompatibility of COF@T.

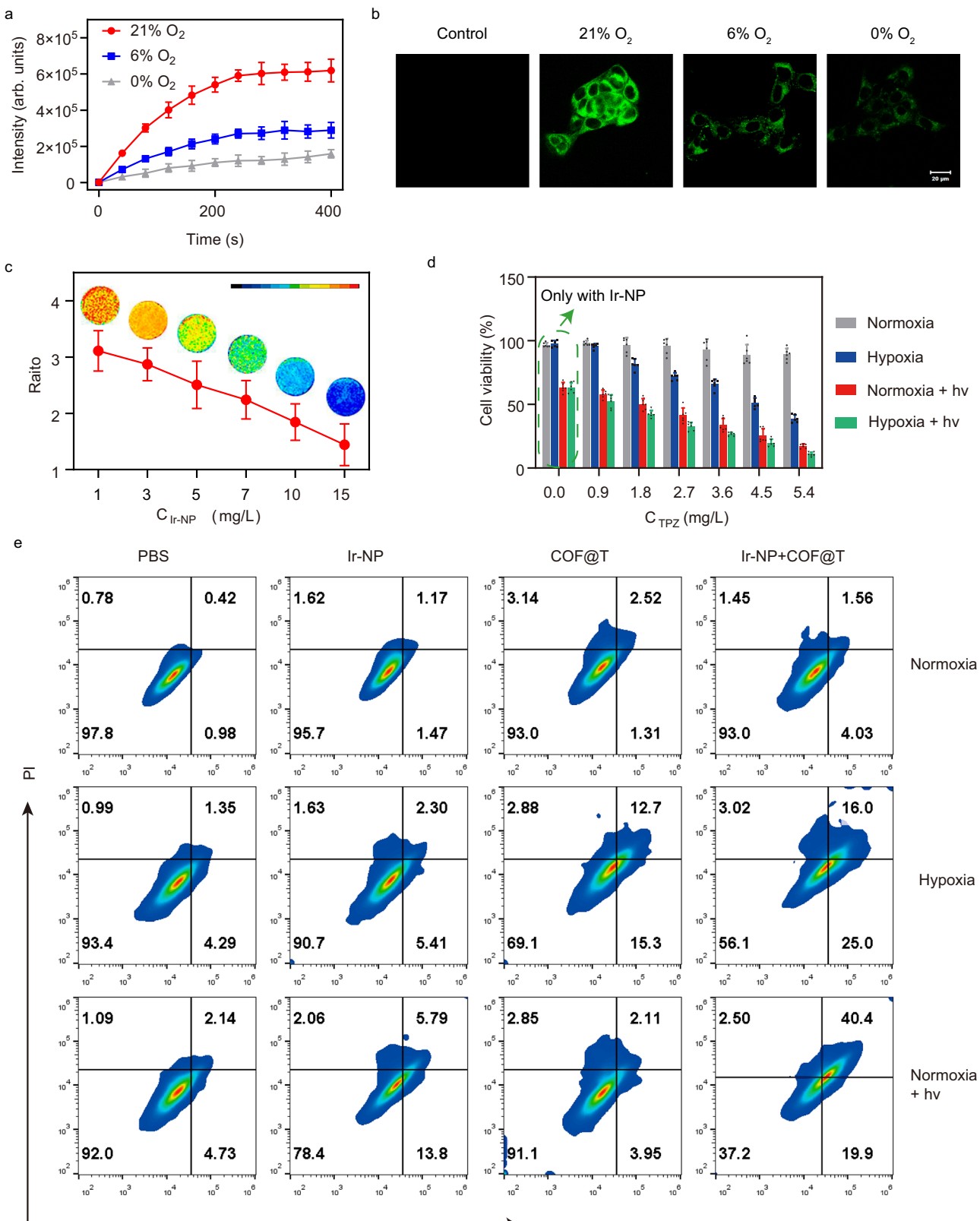

**Fig. 5 | Improving therapeutic effect of TPZ by light-activated modulating hypoxia. a** The fluorescence changes of SOSG with Ir-NP at different oxygen concentration under light irradiation. The statistical data are expressed as mean values ± S.D. ($n = 3$ independent experiments). **b** CLSM images of 4T1 cells after incubation with Ir-NP at different oxygen concentration under light irradiation. ($n$ = three independent experiments). **c** The ratio-metric imaging and results after light irradiation of Ir-NP with different concentration. The statistical data are expressed as mean values ± S.D. ($n = 3$ independent experiments). **d** The cell viabilities of 4T1 cells under different conditions after treatment with Ir-NP and COF@T. The statistical data are expressed as mean values ± S.D. ($n = 6$ independent experiments). **e** Annexin-V/PI labelling analysis for 4T1 cells after different treatments. The term (arb. units) is abbreviated for arbitrary units.

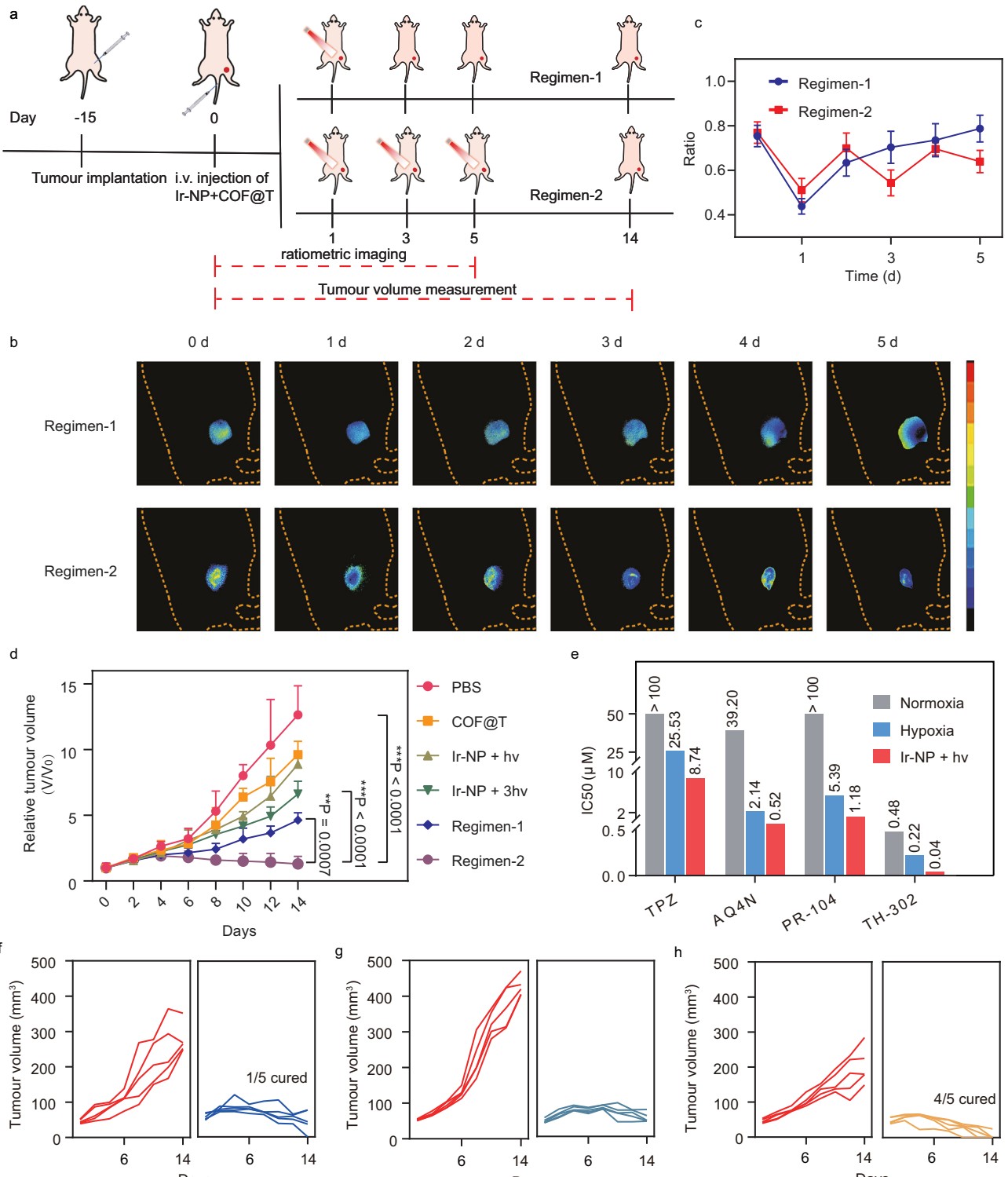

**Fig. 6 | The antitumour effect of Ir-NP based theranostic system in vivo.**
**a** Schematic of Regimen-1 and Regimen-2 with our theranostic system. The ratio-
metric imaging (**b**) and results (**c**) of mice treated with Regimen-1 and Regimen-2.
The statistical data are expressed as mean values ± S.D. ($n = 5$ biologically inde-
pendent animals). **d** The relative tumour volume changes of the subcutaneous 4T1
tumour-bearing mice in different treatments for 14 days. ***$P < 0.0001$ (PBS vs.
Regimen-2), ***$P < 0.0001$ (Ir-NP + 3hv vs. Regimen-2), **$P < 0.001$ ($P = 0.0007$,

Regimen-1 vs. Regimen-2). One-way ANOVA with Dunnett's multiple comparisons
test. The statistical data are expressed as mean values ± S.D. ($n = 5$ biologically
independent animals). **e** IC50 values of four prodrugs under different conditions.
**f–h** The left curves of each figure showing the relative tumour volumes change after
treatment with COF@AQ, COF@PR and COF@TH, respectively, while the right
curves representing the tumour volumes change after treatment with Ir-NP and
prodrugs in combination (refer to Regimen-2).

## Improved antitumour efficacy in other hypoxia-activated prodrugs

To test whether this theranostic combination can promote the antitumour activity of other hypoxia-specific chemotherapeutic drugs, we prepared COF loaded with AQ4N (COF@AQ), PR104 (COF@PR), and TH-302 (COF@TH), respectively, as shown in Supplementary Fig. 32. $IC_{50}$ values of the other three hypoxia-specific drugs under irradiation with Ir-NP were all lower than their values under normoxia and hypoxia conditions (Fig. 6e). When treating mice bearing 4T1 tumours using COF@AQ, COF@PR, COF@TH and Ir-NP in combination with light irradiation (refer to Regimen-2), the tumour growth rates were all significantly inhibited comparing to those treated with COF@X alone (Fig. 6f–h and Supplementary Fig. 33). Notably, 4/5 mice in the group treated with COF@TH and 1/5 mice in group treated with COF@AQ were cured. These results demonstrate that the hypoxia modulation is especially important for hypoxia-activated prodrug in tumour treatment, and our theranostic system combination can significantly enhance the antitumour efficiency of hypoxia-specific drugs.

## Discussion

Combining molecular or functional imaging capabilities with biomarker targeted therapies is a desirable precision medicine approach. Theranostic systems capable of precisely monitoring hypoxia for cancer are very limited. We have constructed the light-activated theranostic combination with a NIR ratiometric probe as the diagnostic moiety and hypoxia-specific prodrugs as the therapeutic agent. We demonstrated that the NIR probe with Ir-NP could provide accurate quantitative ratiometric imaging of tissue oxygenation through its high sensitivity of phosphorescence emission to the changes in the oxygen concentration. On the other hand, it also exerts the PDT effect using highly specific light irradiation while adding antitumour effect of hypoxia-specific prodrugs.

This hypoxia-specific theranostic combination, which was triggered by light and use oxygen to make optical probe and prodrug cooperation, constituted an effective way to combine imaging moiety and treatment agent. Although most PDT agents, including Ir-NP alone, have limited therapeutic effect when the tissue oxygenation level in the tumour becomes low and hypoxic, incorporating hypoxia activatable prodrug-based with Ir-NP resurrects the efficacy as shown in our reported theranostic combination. Our data shows that the reduction in tumour oxygen concentration as the result of the initial PDT can effectively enhance the cytotoxicity of the hypoxia-specific prodrugs. Single injection combined multi-light irradiations can obtain better treatment response than single injection with single irradiation, further highlighting that merit of our theranostic combination, that is, quantitative hypoxia measurement can offer a guide to prepare treatment programme and drug selection. Thus, our theranostic combination provides a strategy to develop hypoxia mediated tumour therapy. In addition, hypoxia is considered as a risk factor to promote tumour metastasis. Following the 14-day treatment regimen, histological examination of major organs in mice using hematoxylin and eosin (H&E) staining did not reveal any evidence of metastasis. Further advancements in the hypoxia-specific theranostic combination may involve the exploration of probes with enhanced sensitivity and specificity for real-time monitoring of tissue oxygenation. Additionally, expanding the repertoire of hypoxia-specific prodrugs incorporated into the system could offer a wider range of therapeutic options for personalised cancer treatment.

## Methods

### Ethical regulations

All animal experiments were performed in compliance with the protocols approved by the Animal Ethical and Welfare Committee at Nanjing University (Nanjing, China) (Protocol Numbers: IACUC-D2303187). In accordance with the requirements, the size of the subcutaneous tumour and body tumour of mice must not exceed 2000 mm$^3$, in which the diameter of any dimension must be <10 mm. Once this size is reached, euthanasia must be performed. In every animal experiment described in this article, the maximal tumour size/burden of the mouse was never exceeded.

### Materials

Phenylacetic acid (99.0%), zinc chloride (98.0%), phthalimide (98.0%), zinc phenylacetate (98.0%), α, ω-amino-terminated polyethylene glycol (PEG, $Mw$ = 2 kDa, 99.5%), chloro (1,5-cyclooctadiene) iridium(I) dimer ([Ir(COD)$_2$Cl$_2$]$_2$, 95.0%) and chlorosulphonic acid (97.0%)were obtained from J & K Chemical Ltd. Tirapazamine (TPZ), AQ4N, PR104, TH302 and solvents were purchased from Sigma-Aldrich Chemical Co. All the commercially available reagents were used as received without any further purification. The cell culture products were purchased from Thermo Fisher Scientific unless otherwise stated.

### Preparation of Ir-NP

The preparation of IrTBP followed a reported method[41–43]. Briefly, the phthalimide (2.94 g, 20.0 mmol), phenylacetic acid (3.60 g, 26.6 mmol) and zinc phenylacetate (1.68 g, 5.0 mmol) were added into a 100 mL Schlenk flask filled with argon. The mixture was heated and stirred at 360 °C for 1 h. The mixture was cooled, dissolved in acetone, precipitated with DI water and washed three times. The crude products were purified on neutral alumina column to obtain ZnTBP. Then, ZnTBP (500.0 mg, 0.57 mmol) was dissolved in 200 mL dichloromethane and 150 mL hydrochloric acid solution (30% HCl). The solution was stirred for 1 h at room temperature. After that, 1000 mL DI water was added into the solution and extracted with dichloromethane (3 × 300 mL). After concentrating the dried organic layer to a minimum volume, the resulting residue was subjected to silica gel column chromatograph to yield H$_2$TBP as dark-green powder. H$_2$TBP (100 mg, 0.12 mmol) and [Ir(COD)$_2$Cl$_2$]$_2$ (125 mg, 0.186 mmol) were dissolved in 80 mL ethylene glycol and stirred at 170 °C for 6 h. After completion of the reaction, 100 mL DI water was added into the mixture. The precipitate was collected and washed three times. The dried crude product was dissolved in acetone and purified by chromatography using a neutral alumina column. The column was eluted with toluene/acetone (1:1, v/v) firstly to remove the impurities and then eluted with acetone/methanol (95:5, v/v) to obtain Iridium meso-tetraphenyltetrabenzoporphyrin (IrTBP).

The above prepared IrTBP (50.0 mg, 0.05 mmol) was added into 7 mL chlorosulfonic acid and stirred overnight at room temperature for 12 h. After completion of the reaction, the mixed solution was added slowly dropwise to 100 mL saturated NaCl solution at -5 °C and filtered quickly. The precipitate was washed with ice water and the resulting IrTBP-chlorosulfonate was dried in an oven with no more purification. Tetraphenylporphyrin (TPP)-chlorosulfonate was prepared with a similar procedure. The fresh prepared IrTBP-chlorosulfonate (10.0 mg) and TPP-chlorosulfonate (40.0 mg) were added into a flask with dichloromethane (10.0 mL). After stirred 5 min and dissolved, the mixture was added into α, ω-amino-terminated polyethylene glycol (PEG, $Mw$ = 2 kDa, 500.0 mg) and pyridine (0.05 mL) as catalytic. After stirred at room temperature for 3 d, the solvent was removed by centrifugation and the residue was added into 30 mL DI water. After 30 min sonication and subsequent filtration, the collect supernatant was dialysed (molecular weight cut off (MWCO) 3500 Da) for one week with DI water and then lyophilised to yield the final product.

### Cell culture

Murine brain microvascular epithelium Bend.3 cells, murine breast cancer 4T1 cells, and murine hepatic cancer H22 cells were purchased from Chinese Academy of Science Cell Bank for Type Culture Collection (Shanghai, China); H22 and Bend.3 cells were incubated at Dulbecco's Modified Eagle Medium (DMEM) supplemented with 10% fetal

bovine serum (FBS) and 1% penicillin/streptomycin at 37 °C in 5% $CO_2$. 4T1 cells were incubated at Roswell Park Memorial Institute (RPMI) 1640 supplemented with 10% fetal bovine serum (FBS) and 1% penicillin/streptomycin at 37 °C in 5% $CO_2$.

## Ratiometric imaging of Ir-NP in vitro

Aqueous solutions of Ir-NP were prepared at 0.1, 0.2, 0.5, 1, 2, 3 and 4 mg/L, respectively. They were placed under three different oxygen levels (0%, 6% and 21% O2, created by Anaero Pack-Anaero). After 30 min, the luminescence images recorded at $675 \pm 25$ nm and $775 \pm 25$ nm were obtained with the imaging system, respectively. The excitation light source was a xenon lamp at 605 nm, and the ratiometric images were calculated using the ImageJ software.

4T1 cells were inoculated in 96-well plates and divided into three groups. 20 μL of Ir-NP solution (50 mg/L) was added to each well and incubated for 4 h. Each well was isolated from air by adding non-toxic oil (Enzo Life Sciences), before the incubation of cells continued for 6 and 12 h, respectively. The luminescence images windows were obtained with the optical in vivo imaging system at $675 \pm 25$ nm and $775 \pm 25$ nm, respectively, using the excitation wavelength of 605 nm. To study the oxygen consumption rate of different cells, H22, 4T1 and Bend.3 cells were inoculated in three groups of 96-well plates, respectively. 20 μL of Ir-NP solution (50 mg/L) was added to each well and incubated for 4 h, and then processed and imaged using the conditions described above.

## Animals

Female BALB/c mice (4 weeks old) were bought from the Animal centre of Drum-tower Hospital. All mice during the experiments were bred in a pathogen-free facility with a 12 h light/dark cycle at 20 ± 3 °C and 40−50% humidity and had libitum access to food and water.

## Ratiometric imaging of tumour-bearing mice using Ir-NP

BALB/c mice bearing 4T1 breast tumours were prepared by injecting 100 μL of 4T1 cells ($5 \times 10^6$) in PBS suspension into the right posterior side. Once tumours grew to approximately 80 mm³, mice were injected with Ir-NP (200 μL, 1 mg/mL) intravenously. The luminescence images windows were obtained with the optical in vivo imaging system at $675 \pm 25$ nm and $775 \pm 25$ nm at different time. After mice were sacrificed, their major organs and tumours were collected and imaged. To study the real-time monitoring oxygen, 100 μL of 4T1 cells ($5 \times 10^6$) in PBS suspension was injected into the left hind leg of BALB/c mice. After the tumours grew to about 40 mm³, Ir-NP (200 μL, 1 mg/mL) was administered intravenously. After 24 h, the left hind leg of mice was ligatured using a bundle to slow down the blood flow. After 10 min, the bundle was released. The luminescence imaging is performed at 5-min intervals throughout the procedure, lasting 30 min in total.

## Biodistribution of Ir-NP

Ir-NPs was intravenously injected into mice with subcutaneous 4T1 tumours. At different time points, mice were sacrificed and the tumours to collect major organs (including heart, liver, spleen, lung, kidney). Three samples were set up at each time point and the saline-injected mice were used as the control group. All the collected mouse tissues were mechanically broken in 10 mL centrifuge tubes respectively and 5 mL dichloromethane was added to each centrifuge tube as extraction solution. Subsequently, the tubes were homogenised thoroughly and extracted for two days at room temperature. After the centrifugation and removal of the precipitate, the concentration of Ir-NP was determined by fluorescence spectrometry (Excitation: 635 nm. Emission: 665 nm).

## Preparation of COF@T

COF@T was prepared with previous report. In brief, benzidine (0.045 mmol, 8.3 mg) and 2,4,6-trihydroxybenzene-1,3,5-tricarbaldehyde (0.030 mmol, 6.3 mg) were dissolved in 100 mL of anhydrous ethanol and then stirred for 1 h at room temperature. The solution was centrifuged at 6000 g for 20 min and the supernatant was removed. Next, the solid was washed three times by dimethylacetamide (DMAc) and dried at vacuum oven. The obtained yellow solid was mixed with TPZ (5 mg) and mPEG-$NH_2$ ($Mw = 5000$, 10 mg) in DMSO (20 mL). After stirred 12 h, the impurities were removed by dialysis and the product was obtained by lyophilization as powder, stored in -5 °C.

## Therapeutic effect of our theranostic system

4T1 cells ($5 \times 10^6$) in PBS suspension was injected into the right posterior side of BALB/c mice to establish subcutaneous tumours. When the tumour volume reached roughly 100 mm3, the mice were divided into five groups (5 mice in each group): (1) PBS; (2) COF@T; (3) Ir-NP + hv; (4) Ir-NP + 3hv; (5) Regimen-1; (6) Regimen-2. All samples were administrated via tail vein, respectively. The dose of TPZ was 9 mg/kg and the dose of Ir-NP was 20 mg/kg. The tumour of mice in (3) and (5) was irradiated with 635 nm laser (150 mW cm-2) for 15 min at 24 h after the injection. And the tumour of mice in (4) and (6) was irradiated with 635 nm laser (150 mW cm-2) for 5 min at 24 h, 48 h and 72 h. The tumour volume and body weight of mice were measured on alternate day. The volume of tumour was calculated as follows:

$$Volume = 0.5 * a * b^2$$

In the equation, a represents the maximum diameter of tumour while b represents the minimum diameter of tumour. On day 14, the mice were sacrificed, and tumour weight was recorded, and histopathological analysis of major organs was performed by H&E staining.

## Statistical analyses

Error bars are reported as mean ± s.d. Differences between groups were compared by analysis of variance (ANOVA) and Student's $t$-test. A $P$-value < 0.05 was considered to be statistically significant.

## Reporting summary

Further information on research design is available in the Nature Portfolio Reporting Summary linked to this article.

# Data availability

All data generated or analysed during this study are included in this published article (and its Supplementary Information files). All other data are available from the corresponding authors upon request.

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

## Acknowledgements

This work was supported the Natural Science Foundation of China (92163214 and 52333003 to X.J.) and the Natural Science Foundation of Jiangsu Province (BK20202002 to X.J.).

## Author contributions

L.G. and X.J. conceived and designed the research. L.G., Y.T., C.W. and J.C. performed the experiments. L.G., H.M. and X.J. analysed the data and wrote the manuscript. X.J. supervised the project. All authors discussed the results throughout the project and approved the final version of the manuscript.

## Competing interests

The authors declare no competing interests.
