## [Peer Review File · Nature Communications]

A Light-Activatable Theranostic Combination for Ratiometric Hypoxia Imaging and Oxygen-Deprived Drug Activity EnhancementREVIEWER COMMENTS

Reviewer #1 (Remarks to the Author):

In this manuscript, Jiang et al. described a light-activatable theranostic nanoplatform. A quantitative probe for hypoxia was constructed using a combination of fluorescence and phosphorescence with different sensitivities to oxygen. The ratio of luminescent intensity at 665 nm to 774 nm exhibited an apparent linear relationship with the oxygen level. The ratio of phosphorescence intensities of 774 nm under the oxygen-free to oxygen-saturated conditions reached 6. On the other hand, consumption of oxygen by photosensitizers, thereby boosting the efficacy of hypoxia-activated chemotherapeutic drugs, is a novel strategy. The theranostic effect was demonstrated in vivo. However, there are some questions listed as follows that need to be addressed.

My suggestion is: accept after major revision.

1. As the hypoxic quantitative probes are constructed from a combination of two molecules, whether the different assembly ratios could affect the detection results should be discussed.
2. The results of single injection combined multi-light irradiations is attractive. The two components (Ir-NP and COF@T) are of different sizes and may differ in their tumor enrichment capacity and blood half-life times. Is the plan for co-administrated intravenously dosing taking this into account? On day 5, the imaging results showed that Ir-NP was still at the tumor site. However, for COF@T, it is questioned whether it is also present in the tumor.
3. The hypoxia-activated chemotherapeutic drugs (TPZ) and photosensitizers are spatially close. Does the highly reactive singlet oxygen generated by photosensitizers disrupt TPZ?
4. TPZ was simply loaded into the COF. Could the authors provide the loading capacity, release rate, and release proportion?
5. Fig. 5, results of 'Hypoxia + hv' should be added for comparison.
6. How did the authors choose to dose the animals with 20 mg/kg for Ir-NP and 9 mg/kg for TPZ?
7. The detailed prodrugs for Fig. 6f-h should be presented in the caption.

Reviewer #2 (Remarks to the Author):

Here the authors have studied the theranostic potential of near-infrared (NIR)-emitting ratiometric phosphorescence probe (Ir-NP) in combination with a prodrug-loaded covalent organic framework (COF@T). But the material synthesis and PDT application of IrTBP is not novel at all.

-The authors evaluated only the cytotoxicity of the synthesized material in the presence of a particular wavelength of light by Annexin-FITC. Authors didn't even discussed why the synthesized material is showing anticancer activity?

-There is no positive control.

-Authors didn't show any healthy cell line data. It might kill the healthy cell line. There is no data.

-The Discussion portion is too small. Discussion part is not justified at all for a journal like Nature Communication. Thus, I recommend rejection to this manuscript.

Reviewer #3 (Remarks to the Author):

The work by Ge et al. designed a theranostic combination of light-activated ratiometric hypoxia imaging and hypoxia modulating and prodrug activation. Totally, it is an exquisitely designed combination that can realize mutual benefit. The manuscript is logically reasonable and well-organized with sufficient data to support their viewpoints. I have the following considerations and suggestions before possible publication of the paper.

1. Ir-NP is self-assembled from several small molecules. Although the authors have evaluated the pH-resistant durability with the pH range from 2 to 12, I worry about the stability of the nanoparticle under physiological and tumor condition, especially under light irradiation. If it is not stable, the ratiometric hypoxia imaging will be questionable.
2. Would Ir-NP bring some photothermal effect under the 650 nm light?
3. For the in vivo imaging of hypoxia and pharmacokinetics, the author only used fluorescent imaging to get all the results. I think some other quantitative characterization such as ICP (using Ir) can be used to support the imaging results.
4. Hypoxia is considered as a risk factor to promote tumor metastasis. So, is there any risk of activation of the prodrugs through enhancing enhanced hypoxia microenvironment of tumour? The author should discuss it.
5. Please supply the physicochemical properties of COF@T, such as size distribution, surface chemistry, porous structure, so that the loading and release of drugs can be validated.
6. The authors found hepatic cancer H22 cells exhibiting more rapid oxygen consumption than 4T1 cells. How about the combination therapeutic effect on H22 tumors on animals?

Response to Reviewer's Concerns

We are grateful to the reviewers for their insightful comments and constructive critiques as well as valuable suggestions. We have revised our manuscript, including providing additional clarifications and discussions and adding new data/results to address these points accordingly.

Reviewer #1 (Remarks to the Author):

In this manuscript, Jiang et al. described a light-activatable theranostic nanoplatfrom. A quantitative probe for hypoxia was constructed using a combination of fluorescence and phosphorescence with different sensitivities to oxygen. The ratio of luminescent intensity at 665 nm to 774 nm exhibited an apparent linear relationship with the oxygen level. The ratio of phosphorescence intensities of 774 nm under the oxygen-free to oxygen-saturated conditions reached 6. On the other hand, consumption of oxygen by photosensitizers, thereby boosting the efficacy of hypoxia-activated chemotherapeutic drugs, is a novel strategy. The theranostic effect was demonstrated in vivo. However, there are some questions listed as follows that need to be addressed.

My suggestion is: accept after major revision.

Q1. *As the hypoxic quantitative probes are constructed from a combination of two molecules, whether the different assembly ratios could affect the detection results should be discussed.*

Response: We thank the reviewer's comment. We investigated the effect of emission spectrum for different ratios of IrTBP/TPP, and the corresponding data are presented in **Supplementary Table 1**. Our analysis revealed that when the mass ratio of IrTBP/TPP varied from 1/20 to 1/1, the ratio of I_{665} / I_{774} also changed from 51.22 to 0.15. However, ratios that are too large or too small are not suitable for subsequent calculations. Therefore, we determined the optimal mass ratio to be 1/4, where the ratio of I_{665}/I_{774} was 1.00. To clarify this point, we have added the following sentence "The mass ratio

of chlorosulfonylated IrTBP to chlorosulfonylated TPP was determined as 1/4 to ensure that the ratio of luminescent intensity at 665 nm to 774 nm (I_{665}/I_{774}) was suitable for detection (Supplementary Table 1).” at the page 5, line 11 of revision.

Q2. The results of single injection combined multi-light irradiations is attractive. The two components (Ir-NP and COF@T) are of different sizes and may differ in their tumor enrichment capacity and blood half-life times. Is the plan for co-administrated intravenously dosing taking this into account? On day 5, the imaging results showed that Ir-NP was still at the tumor site. However, for COF@T, it is questioned whether it is also present in the tumor.

Response: Owing to the nano structure of Ir-NP and COF@T, both could effectively delivery to the tumor passively. Although their half-life times are not identical, they both have high concentrations in the tumor throughout the treatment process, thus ensuring their combined therapeutic effects. We measured the changes of COF@T concentration at the tumor throughout the treatment process. The results were shown as **Supplementary Fig. 26**. It can be seen that on day 6, there is still a certain drug concentration, thus could confirm the successful implementation of our light-activatable theranostic combination. We have added the following sentence “The concentration changes of TPZ at the tumor site of mice from two groups during the 6-day regimen are shown in Supplementary Fig. 26. Notably, the result suggests that TPZ consistently maintains a high concentration throughout the duration of treatment, thereby ensuring the successful implementation of therapeutic strategy.” at the page 10, line 17 of revision.

Q3. The hypoxia-activated chemotherapeutic drugs (TPZ) and photosensitizers are spatially close. Does the highly reactive singlet oxygen generated by photosensitizers disrupt TPZ?

Response: We thank the reviewer for this comment. tirapazamine (TPZ) is a benzotriazine compound that exhibits selective cytotoxicity towards hypoxic cells. Its

cytotoxic effects are attributed to its metabolism by intracellular reductases under hypoxic conditions, which leads to the formation of cytotoxic benzotriazinyl radicals and hydroxyl radicals. The mechanism of TPZ is shown in the following figure. The formation of radicals is facilitated by higher concentrations of singlet oxygen and lower concentrations of oxygen. We also cite this reference (*Journal of the American Society for Mass Spectrometry* **14**, 881-892, (2003).) as Ref. 24 in revision.

Scheme 1

The mechanism of radicals generation by TPZ

Q4. TPZ was simply loaded into the COF. Could the authors provide the loading capacity, release rate, and release proportion?

Response: The loading capacity of TPZ was determined by measuring its absorption at a wavelength of 471 nm. We have added the following sentence “The quantity of loaded drug was determined using a UV-vis spectrophotometer at a wavelength of 461 nm. The drug loading capacity were calculated using the following equations:

$$\text{Loading capacity} = \frac{\text{Weight of drugs in NPs}}{\text{Weight of NPs}} \times 100\%” \text{ in the Supporting information.}$$

We tested the release of COF@T over a period of 5 days using a dialysis approach. At 24 h, about 27% of loaded TPZ is released from the COF@T. At the end of the release profile (5 day), this value reaches 35.1%. We have added the results as **Supplementary Fig. 21**.

To clarify these two points, we have added the following sentence “The loading capacity of TPZ in COF@T was calculated as 12.1% and the TPZ release from COF@T was shown as Supplementary Fig. 21.” at the page 9, line 10 of revision.

Q5. Fig. 5, results of 'Hypoxia + hv' should be added for comparison.

Response: The results of 'hypoxia + hv' was added into Fig. 5d and e in the revision. The results revealed that the simultaneous hypoxia and light irradiation could significantly enhance the cytotoxicity of the combined treatment with Ir-NP and COF@T. This combined approach yielded an augmented therapeutic effect compared to individual treatments, highlighting its potential as an effective treatment strategy.

Q6. How did the authors choose to dose the animals with 20 mg/kg for Ir-NP and 9 mg/kg for TPZ?

Response: Ir-NP served a dual function in the system, encompassing both imaging and PDT. The dose of 20 mg/kg for Ir-NP was chosen to ensure the imaging results and therapeutic effect. As for TPZ, we refer to the literatures and decide to use the dose (*Advanced Materials* **31**, 1805955, (2019). *Nano Letters* **21**, 3218-3224, (2021).). The histological examination was performed after the treatment and the results proved that our therapeutic system didn't have any obviously damage to the tissue of mice.

Q7. The detailed prodrugs for Fig. 6f-h should be presented in the caption.

Response: We have added the following sentence "The left curve of each figure showing the relative tumour volumes change after treatment with COF@AQ, COF@PR and COF@TH, respectively, while the right curve presenting the tumour volume change after treatment with Ir-NP and prodrugs in combination (refer to Regimen-2)." in the caption of Fig. 6 of revision.

Reviewer #2 (Remarks to the Author):

Q1. Here the authors have studied the theranostic potential of near-infrared (NIR)-emitting ratiometric phosphorescence probe (Ir-NP) in combination with a prodrug-loaded covalent organic framework (COF@T). But the material synthesis and PDT application of IrTBP is not novel at all.

Response: We thank the reviewer for the comments. In this work, we used IrTBP as a functional small model molecule and combined it with tetraphenylporphyrin to assemble nanoparticles (Ir-NP) which could ratiometrically image to hypoxia and enhance photodynamic therapy (PDT) efficiency in conjunction with COF@T in a hypoxic environment. The focus of this work lies in utilizing the PDT of IrTBP to consume oxygen within tumor cells, thereby enhancing the efficacy of the cytotoxicity of hypoxia prodrug TPZ. Here Ir-NP plays two roles, including quantitative monitoring of oxygen concentration by ratiometric imaging and consuming the oxygen of tumour under light excitation by PDT. Meanwhile, the enhanced hypoxia microenvironment of tumour can raise the cytotoxicity of prodrug loaded in COF, resulting in boosting antitumour therapeutic effects in vivo. Moreover, the treatment of single injection combined multi-light irradiations has important clinic significance. Thus, this study introduces a novel method for the theranostic of tumor hypoxia which is new and important in biomedical area.

Q2. The authors evaluated only the cytotoxicity of the synthesized material in the presence of a particular wavelength of light by Annexin-FITC. Authors didn't even discuss why the synthesized material is showing anticancer activity?

Response: Annexin-FITC flow cytometry is a common technique that enables the detection and quantification of cell apoptosis by specifically binding to phosphatidylserine exposed on the outer surface of apoptotic cells. In normal cells, phosphatidylserine is predominantly located on the inner side of the cell membrane, preventing Annexin-FITC binding. However, during apoptosis, the membrane undergoes changes, resulting in the externalization of phosphatidylserine, allowing Annexin-FITC binding. Furthermore, to demonstrate the anticancer efficacy of our synthesized material, we employed both MTT assay and flow cytometry. Both methods yielded consistent results, confirming the anticancer effects of the material.

Ir-NP is mainly composed of porphyrin derivatives. There are many literatures which have proved that the porphyrin derivatives can generate cytotoxic ROS by

photochemical reaction under the irradiation of corresponding light. To clarify this point, we cite literatures (*Chemical Society Reviews* **40**, 340-362, (2011). *Nature Communications* **9**, 3653, (2018). *ACS Nano* **14**, 13569-13583, (2020).) as Ref. 35-37.

Q3. *There is no positive control.*

Response: We added the MTT and flow cytometry experiments results under ‘hypoxia + hv’ as positive control. The results were shown as **Figure 5d and 5e**. The results revealed that the simultaneous hypoxia and light irradiation could significantly enhance the cytotoxicity of the combined treatment with Ir-NP and COF@T. This combined approach yielded an augmented therapeutic effect compared to individual treatments, highlighting its potential as an effective treatment strategy.

Q4. *Authors didn't show any healthy cell line data. It might kill the healthy cell line. There is no data.*

Response: We test the viability of Bend.3 cells (a healthy cell line) incubated with our Ir-NP and TPZ theranostic combination by MTT assay. The results showed that our theranostic combination didn't have noticeable cytotoxicity. We have added the results as **Supplementary Fig. 23** and the sentence “Reassuringly, the theranostic combination exhibited remarkably low cytotoxicity to healthy cells under normoxia conditions without light irradiation (Supplementary Fig. 23).” at the page 9, line 16.

Q5. *The Discussion portion is too small. Discussion part is not justified at all for a journal like Nature Communication.*

Response: We have added one paragraph discussion at the page 12, line 15-22 of revision.

Reviewer #3 (Remarks to the Author):

The work by Ge et al. designed a theranostic combination of light-activated ratiometric

hypoxia imaging and hypoxia modulating and prodrug activation. Totally, it is an exquisitely designed combination that can realize mutual benefit. The manuscript is logically reasonable and well-organized with sufficient data to support their viewpoints. I have the following considerations and suggestions before possible publication of the paper.

Q1. *Ir-NP is self-assembled from several small molecules. Although the authors have evaluated the pH-resistant durability with the pH range from 2 to 12, I worry about the stability of the nanoparticle under physiological and tumor condition, especially under light irradiation. If it is not stable, the ratiometric hypoxia imaging will be questionable.*

Response: We thank the reviewer's comment. To prove the stability of the nanoparticles under physiological and tumor condition, we measured the particle size before and after light exposure in reducing and hypoxia microenvironments by DLS, respectively. The results showed that there was no significant change in the size of the nanoparticles, which proved the stability of the nanoparticles. The results were shown as **Supplementary Fig. 7**.

Q2. *Would Ir-NP bring some photothermal effect under the 650 nm light?*

Response: To evaluate the photothermal effect of Ir-NP, we subjected its solution to laser irradiation (650 nm, 0.30 mW/cm²) for a period of 10 minutes and measured the resulting temperature change. Our observations indicate that the solution temperature remained relatively stable during the treatment, most likely owing to the low intensity of the applied light.

Q3. *For the in vivo imaging of hypoxia and pharmacokinetics, the author only used fluorescent imaging to get all the results. I think some other quantitative characterization such as ICP (using Ir) can be used to support the imaging results.*

Response: We thank the reviewer's valuable suggestion. we used ICP-AES method to determine the Ir concentration in the major organs and the tumors of mice. The results were shown as **Supplementary Fig. 14**.

Q4. Hypoxia is considered as a risk factor to promote tumor metastasis. So, is there any risk of activation of the prodrugs through enhancing enhanced hypoxia microenvironment of tumour? The author should discuss it.

Response: Even though hypoxia is known to promote tumor metastasis, the use of TPZ in this study effectively induced free radical formation under hypoxic conditions, leading to the destruction of cancer cells and inhibition of tumor growth and metastasis. Following the 14-day treatment regimen, histological examination of major organs in mice using hematoxylin and eosin (H&E) staining did not reveal any evidence of metastasis. We have added the discussion in Page 12, line 15 of revision.

Q5. Please supply the physicochemical properties of COF@T, such as size distribution, surface chemistry, porous structure, so that the loading and release of drugs can be validated.

Response: We thank the reviewer's valuable suggestion. The size distribution of COF@T was determined by TEM and DLS. The results were shown as **Supplementary Fig. 19** of revision. And we proved the porous structure of COF by BET experiments. The results were shown as **Supplementary Fig. 20**. We added the following sentence "The BET surface area measurement of COF was found to be 580.42 m² g⁻¹, indicating a highly porous structure. Additionally, the pore size of COF was determined to be 2.4 nm (**Supplementary Fig. 20**)." at the page 9, line 7 of revision.

The loading capacity of TPZ was determined by measuring its absorption at a wavelength of 471 nm. We have added the following sentence "The quantity of loaded drug was determined using a UV-vis spectrophotometer at a wavelength of 461 nm. The drug loading capacity were calculated using the following equations:

$$\text{Loading capacity} = \frac{\text{Weight of drugs in NPs}}{\text{Weight of NPs}} \times 100\%$$
 in supporting information.

We tested the release of COF@T over a period of 5 days using a dialysis approach. At 24 h, about 27% of loaded TPZ is released from the COF@T. At the end of the release profile (5 day), this value reaches 35.1%. We have added the results as

Supplementary Fig. 21.

We have added the following sentence “The loading capacity of TPZ in COF@T was calculated as 12.1% and the TPZ release from COF@T was shown as **Supplementary Fig. 21.**” at the page 9, line 10 of revision.

Q6. The authors found hepatic cancer H22 cells exhibiting more rapid oxygen consumption than 4T1 cells. How about the combination therapeutic effect on H22 tumors on animals?

Response: We used Ir-NP to find that H22 cells have a greater capacity for oxygen consumption compared to 4T1 cells; therefore, it is reasonable to speculate that oxygen consumption is likely to be greater and oxygen concentration lower in H22 tumors than in 4T1 tumors. And our light-activatable theranostic combination is needed to consume the oxygen under the light irradiation to perform the photodynamic therapy and enhance the toxicity of TPZ, thus achieving a combined therapeutic effect. However, for H22 tumors, its oxygen concentration is much lower than that of 4T1 tumors, thus making it difficult to further enhance the toxicity of TPZ and not fully reflecting the advantages of our light-activatable theranostic combination.

REVIEWER COMMENTS

Reviewer #1 (Remarks to the Author):

The author has revised the manuscript well according to the questions and suggestions raised by reviewers.

Reviewer #2 (Remarks to the Author):

Authors have made necessary revisions and the manuscript looks fine now. Thus, I am recommending this revised version for publication.

Reviewer #3 (Remarks to the Author):

The authors have response to most of the proposed questions previously. Regarding my previous comments, I still have one minor question. The authors speculate that oxygen consumption is likely to be greater and oxygen concentration lower in H22 tumors than in 4T1 tumors. Could you provide some evidences?

Response to Reviewer's Concerns

We are grateful to the reviewers for their positive comments and valuable suggestions.

We have further revised our manuscript to address reviewers' comments.

Reviewer #1 (Remarks to the Author):

The author has revised the manuscript well according to the questions and suggestions raised by reviewers.

Response: We thank the reviewer for the positive comments.

Reviewer #2 (Remarks to the Author):

Authors have made necessary revisions and the manuscript looks fine now.

Thus, I am recommending this revised version for publication.

Response: We thank the reviewer for the positive comments.

Reviewer #3 (Remarks to the Author):

The authors have response to most of the proposed questions previously. Regarding my previous comments, I still have one minor question. The authors speculate that oxygen consumption is likely to be greater and oxygen concentration lower in H22 tumors than in 4T1 tumors. Could you provide some evidences?

Response: We thank the reviewer for the positive comments. Firstly, from the ratio imaging results in Figure 3c and Figure 3d of the manuscript, it can be observed that, for the same incubation time and conditions, the

imaging ratio (Ch1/Ch2) of H22 cells is lower than that of 4T1 cells. This implies that H22 cells exhibit a stronger oxygen consumption capacity compared to 4T1 cells. Further, to definitively show the differences in hypoxia levels between H22 and 4T1 tumors, we subsequently compared the oxygen concentrations between these two xenografted tumors using oxygen concentration measurement electrodes (gold standard) and pimonidazole staining. Additionally, we employed immunohistochemical staining to compare the concentrations of hypoxia inducible factor 1 α (HIF-1 α) within both tumors. The results are shown in the following figure. Our results indicates that the oxygen concentration in H22 tumors is significantly lower than that in 4T1 tumors, while the concentration of HIF-1 is significantly higher in H22 tumors compared to that of 4T1 tumors. Therefore, we conclude that the hypoxia level is stronger in H22 tumors compared to 4T1 tumors.

This observation is also consistent with previous work where the expression level of the HIF- α in H22 cells is higher than that in 4T1 cells under the same condition (*Plos one* 4 (10): e7629).

Figure R1. (a) The oxygen concentration of H22 and 4T1 tumours measured by oxygen concentration measurement electrodes. Frozen sections of two tumours stained for pimonidazole (b) and HIF-1 α (c).

The data of oxygen concentration measurements for 4T1 and H22 tumors, as well as the immunohistochemical results related to hypoxia, have been included in the Supplementary Information (SI). We also added the sentences “Subsequently, we substantiated the significantly higher degree of hypoxia in H22 xenografted tumour compared to 4T1 xenografted tumour through the utilization of oxygen concentration measurement electrodes and pimonidazole staining (Supplementary Fig. 16). The results of hypoxia-inducible factor-1 α (HIF-1 α) immunohistochemistry staining were also in accordance with this finding (Supplementary Fig. 17).” at page 9, line 12-17 of the revision.

REVIEWERS' COMMENTS

Reviewer #3 (Remarks to the Author):

I have no additional comment and believe the manuscript is acceptable for publication as is.